# Gender Difference in Lithium-Induced Sodium Current Dysregulation and Ventricular Arrhythmogenesis in Right Ventricular Outflow Tract Cardiomyocytes

**DOI:** 10.3390/biomedicines10112727

**Published:** 2022-10-28

**Authors:** Ching-Han Liu, Yao-Chang Chen, Yen-Yu Lu, Yung-Kuo Lin, Satoshi Higa, Shih-Ann Chen, Yi-Jen Chen

**Affiliations:** 1Division of Cardiology, Department of Internal Medicine, Kaohsiung Armed Forces General Hospital, Kaohsiung 80284, Taiwan; 2Division of Cardiology, Department of Internal Medicine, Tri-Service General Hospital, National Defense Medical Center, Taipei 11490, Taiwan; 3Heart Rhythm Center, Division of Cardiology, Department of Medicine, Taipei Veterans General Hospital, Taipei 11217, Taiwan; 4Cardiovascular Research Center, Wan-Fang Hospital, Taipei Medical University, Taipei 11696, Taiwan; 5Department of Biomedical Engineering, National Defense Medical Center, Taipei 11490, Taiwan; 6Division of Cardiology, Sijhih Cathay General Hospital, Sijhih, New Taipei City 22174, Taiwan; 7School of Medicine, Fu-Jen Catholic University, New Taipei City 24257, Taiwan; 8Division of Cardiology, Department of Internal Medicine, School of Medicine, College of Medicine, Taipei Medical University, Taipei 11042, Taiwan; 9Cardiac Electrophysiology and Pacing Laboratory, Division of Cardiovascular Medicine, Makiminato Central Hospital, Urasoe 901-2131, Okinawa, Japan; 10Cardiovascular Center, Taichung Veterans General Hospital, Taichung 40705, Taiwan; 11Department of Post-Baccalaureate Medicine, College of Medicine, National Chung Hsing University, Taichung 40227, Taiwan; 12Graduate Institute of Clinical Medicine, College of Medicine, Taipei Medical University, Taipei 11042, Taiwan

**Keywords:** Brugada syndrome, gender, lithium intoxication, sodium current dysregulation, right ventricular outflow tract

## Abstract

Lithium intoxication induces Brugada-pattern ECG, ventricular arrhythmia, and sudden death with the predominant preference for the male over the female gender. This study investigated the mechanisms of gender difference in lithium-induced arrhythmogenesis. The ECG parameters were recorded in male and female rabbits before and after the intravenous administration of lithium chloride (LiCl) (1, 3, 10 mmol/kg). Patch clamps were used to study the sodium current (I_Na_) and late sodium current (I_Na-late_) in the isolated single male and female right ventricular outflow tract (RVOT) cardiomyocytes before and after LiCl. Male rabbits (*n* = 9) were more prone to developing lithium-induced Brugada-pattern ECG changes (incomplete right bundle branch block, ST elevation and QRS widening) with fatal arrhythmia (66.7% vs. 0%, *p* = 0.002) than in female (*n* = 7) rabbits at 10 mmol/kg (but not 1 or 3 mmol/kg). Compared to those in the female RVOT cardiomyocytes, LiCl (100 μM) reduced I_Na_ to a greater extent and increased I_Na-late_ in the male RVOT cardiomyocytes. Moreover, in the presence of ranolazine (the I_Na-late_ inhibitor, 3.6 mg/kg iv loading, followed by a second iv bolus 6.0 mg/kg administered 30 min later, *n* = 5), LiCl (10 mmol/kg) did not induce Brugada-pattern ECG changes (*p* < 0.005). The male gender is much predisposed to lithium-induced Brugada-pattern ECG changes with a greater impact on I_Na_ and I_Na-late_ in RVOT cardiomyocytes. Targeting I_Na-late_ may be a potential therapeutic strategy for Brugada syndrome-related ventricular tachyarrhythmia.

## 1. Introduction

Lithium is the agent commonly used to treat bipolar disorder. However, cardiac adverse effects, including both benign electrocardiographic (ECG) changes and near-fatal arrhythmias, have been reported [1]. Lithium’s effect on the electrophysiological characteristics of the heart is considered related to its concentration-dependent inhibition of cardiomyocyte voltage-gated sodium (Na^+^) channels which decreases intracellular potassium (K^+^), thus causing ventricular electrical instability [2]. Some lithium-treated patients have experienced cardiac syncope or sudden cardiac death, which was found to be associated with Brugada-pattern ECG changes [3,4].

Lithium has been proven to unmask type 1 Brugada-pattern ECG changes, which was suggested to be caused by its effect on Na^+^ channel inhibition, and the electrical changes imposed by lithium are determined by the duration of lithium treatment and serum lithium concentration [1]. The normalization of ECG or conversions to different Brugada-pattern ECG changes could be seen after the cessation of lithium [5]. Lithium administration has been identified to cause Brugada-pattern ECG changes since 2005, and the majority (71.4%) of the reported cases were male [4,6]. In the drug-induced Brugada syndrome, the male gender was associated with a higher rate of ventricular arrhythmia inducibility than females [7]. As in the drug-induced Brugada syndrome, males were associated with a higher risk of arrhythmic events and a higher rate of ventricular fibrillation inducibility in the channelopathy-induced Brugada syndrome [8]. 

The lithium-related Brugada-pattern ECG changes are notable for a coved ST segment, J point elevation, and T-wave inversion in the right precordial leads V1 and/or V2 [9,10]. The close anatomical proximity of leads V1 and V2 to the right ventricular outflow tract (RVOT)reflects the origin of Brugada syndrome [11]. The abnormal ECG activation in the Brugada syndrome can be temporo-spatially correlated to RVOT by a new CineECG method [12]. Ventricular arrhythmias commonly originate from the RVOT with its distinct electrophysiological characteristics and greater calcium (Ca^2+^) content [13]. The structural abnormalities in the RVOT, which include focal wall thinning or fatty replacement, and localized wall bulging, were found in a large percentage of patients with ventricular arrhythmias [14]. There are abnormalities in the right ventricular (RV) transmembrane activation and repolarization of patients with Brugada syndrome, as regards activation time, repolarization time, and activation recovery intervals [15]. In patients with Brugada syndrome, low-voltage areas, or areas with abnormal electrograms, were commonly identified on the RVOT. Epicardial ablation of the provoked substrate in the RVOT in Brugada patients can eliminate the Brugada syndrome phenotype, which is associated with freedom from ventricular arrhythmia inducibility [16]. 

There are different ionic profiles and Ca^2+^ handlings in the cardiomyocytes between the sexes [17]. Male predominance in electrical changes and the phenotypic expression of Brugada syndrome was related to action potential (AP) parameters and ionic currents in RV epicardium [18]. Gender-specific differences in the late Na^+^ current (I_Na-late_) have been reported in mice ventricular cells, which was translated into effects on cardiac AP durations [19]. Our previous study showed that sex hormones are related to the pathogenesis of ventricular tachycardia originating from the RVOT [20]. However, mechanisms for the gender difference in lithium-induced arrhythmogenesis remain unclear. Therefore, our research aims were to explore the ECG presentations, especially when accompanied by the presence of the Brugada pattern, caused by lithium intoxication, to correlate the ECG changes with the electrophysiological characteristics of RVOT under lithium intoxication, and to find out the potential mechanisms underlying the male preponderance in lithium-induced Brugada-pattern ECG changes. 

## 2. Materials and Methods

### 2.1. Ethics Statement

All the experiments involving animals absolutely complied with the Guide for the Care and Use of Laboratory Animals and were approved by a local ethics review committee (Approval no. IACUC-20-344), National Defense Medical Center, Taiwan. 

### 2.2. Animal Preparations and ECG Recordings

Rabbits of both sexes, which weighed between 2.0 and 2.5 kg, were anesthetized by the inhalation of 2.0–2.5% isoflurane. At least 30 min were allowed for the rabbits’ hemodynamics to stabilize under continuous ECG monitoring. The limb leads were attached as described in the literature [21], and the precordial leads (V1–6) were placed across the rabbits’ anterior thorax with modified positions to detect the right and left heart electrical activities (Figure 1). After a period of acclimatization, 10 s of 12-lead ECG recordings were obtained at the standard 1 mV/10 mm calibration from each rabbit, using the PageWriter TC30 Cardiograph (Philips Medical System, Andover, MA, USA) before and after the lithium chloride (LiCl) solution (1.0, 3.0, and 10.0 mmol/kg), which was sequentially administered with continuous intravenous infusion for 30 min, with a paper speed of 25 mm/sec. QT intervals were measured from the earliest QRS complex onset to the end of the T wave, the latter of which was determined by plotting a tangent extending from the steepest portion of the T wave downslope to the point that it crossed the T-P segment. Both QT and RR intervals were averaged over three consecutive complexes during the sinus rhythm. According to Bazett’s formula, QT intervals were corrected for heart rate (QTc) [22], and a prolonged QTc interval was defined as a QTc interval ≥ 500 ms. To study the therapeutic potential of I_Na-late_ inhibition on LiCl-induced Brugada-pattern ECG changes and arrhythmogenesis, LiCl (10 mmol/kg) was administered in male rabbits 5 min after being treated with ranolazine (3.6 mg/kg intravenous bolus, followed by a second intravenous bolus 6.0 mg/kg 30 min later) [23]. The Brugada-pattern ECG changes noted in this study were defined as the type 1 pattern, i.e., a prominent cove-shaped ST-segment elevation > 2 mm at its peak in the right precordial leads (V1–3), or type 2 pattern, i.e., a saddle-back ST segment with ≥ 2 mm J-point elevation and ≥ 0.5 mm elevation of the terminal ST segment with a positive or biphasic T wave [24].

### 2.3. Isolation of Single RV Epicardial Cardiomyocytes and Patch-Clamp Technique

Rabbits that weighed 2.0–2.5 kg of both sexes were anesthetized with overdosing isoflurane (5% in oxygen) delivered from a precision vaporizer. We confirmed the adequacy of the anesthesia by the lack of corneal reflex and by monitoring the response to painful stimuli elicited by a scalpel tip. Single RVOT cardiomyocytes were enzymatically dissociated from the rabbits as previously described [25]. Briefly, we performed a mid-line thoracotomy to remove the heart and lungs. In these healthy rabbits naive to lithium, the hearts removed under anesthetization were mounted on a Langendorff system to be super-fused antegradely with 95% oxygenated normal Tyrode’s solution at 37 °C, which contained the following (in mM; buffered to pH 7.4 with NaOH): NaCl 137, KCl 5.4, HEPES 10, CaCl_2_ 1.8, MgCl_2_ 0.5, and glucose 11. All traces of blood being washed out, the hearts were perfused for 8–12 min with oxygenated Ca^2+^-free Tyrode’s solution containing collagenase (type I, 300 units/mL) (Sigma Chemical, St. Louis, MO, USA) and protease (type XIV, 0.25 units/mL) (Sigma Chemical, St. Louis, MO, USA), for enzymatic dispersion to take place [26]. Then, the RVOT issues were excised into tiny pieces and lightly shaken with 50 mL of Ca^2+^-free oxygenated Tyrode’s solution until the isolation of single cardiomyocytes. Next, the solution was gradually switched to oxygenated normal Tyrode’s solution.

As described previously, the whole-cell patch clamp experiment was performed in the isolated RVOT cardiomyocytes by using an Axopatch 200B amplifier (Axon Instruments, Foster City, CA, USA) at 35 ± 1 °C. Borosilicate glass electrodes (o.d., 1.8 mm) with a tip resistance of 3–5 MΩ were used. Before the formation of the membrane-pipette seal, the tip potentials were zeroed in Tyrode’s solution. The junction potentials between the bath and pipette solution (9 mV) were corrected for the AP recordings. The resting membrane potential (RMP) was measured during the period from the last repolarization to the next AP. The amplitude of AP (APA) was defined from the RMP to the peak of depolarization. The AP duration (APD) obtained at 90%, 50%, and 20% repolarization of APA was defined as APD_90_, APD_50_, and APD_20_, respectively.

The sodium current (I_Na_) was recorded by using 40-msec. pulses from a holding potential of −120 mV to the test potentials, which varied between −80 and 0 mV in 10 mV increments at a frequency of 3 Hz at room temperature (25 ± 1 °C). The external solution contained (in mM): CsCl 133, NaCl 5, HEPES 5, glucose 5, MgCl_2_ 2, CaCl_2_ 1.8, and nifedipine 0.002 (pH 7.3). Micropipettes were filled with a solution containing (in mM) CsCl 133, TEACl 20, EGTA 10, NaCl 5, MgATP 5, and HEPES 5 (pH 7.3 with CsOH). The I_Na-late_ was recorded with an external solution containing (in mM, at room temperature): NaCl 140, CsCl 5, HEPES 5, MgCl_2_ 2, CaCl_2_ 1.8, nicardipine 0.002, and glucose 5. The amplitude of the I_Na-late_ was measured at a voltage of −20 mV as the mean value between 200 and 250 ms after the membrane depolarization by a 2000-ms. pulse from −140 mV stepped to −20 mV. 

### 2.4. Statistical Analysis

All electrophysiology data are expressed as the mean ± standard error of measurement. A one-way or two-way repeated-measures analysis of variance (ANOVA) (post hoc analysis with Bonferroni correction) or an unpaired t-test was used for comparing differences between the RVOT cardiomyocytes of different groups. The effects of LiCl on ionic currents were compared using the Mann–Whitney rank-sum test or unpaired *t*-test according to the outcome of the normality test. The Pearson chi-square test was used to compare the incidences of fatal arrhythmia. A *p*-value <0.05 was considered statistically significant.

## 3. Results

### 3.1. ECG Findings and Cardiac Arrhythmias in LiCl-Treated Rabbits

At the baseline, there were no significant differences in the heart rate, QT intervals, QTc intervals, P wave amplitudes, or T wave amplitudes between the sexes. LiCl of 1 and 3 mmol/kg did not significantly change the ECG parameters in male and female rabbits. The QRS duration at 10 mmol/kg of LiCl was significantly longer in male rabbits than in female rabbits (Table 1 and Figure 2A). No significant change in QRS duration was noted in female rabbits before and after LiCl administration. In addition, LiCl of the 10 mmol/kg induced type 1 Brugada-pattern ECG changes in six of the nine (67%) male rabbits and in zero female rabbits (Figure 2B). There were no significant differences between the male rabbits with provokable and non-provokable Brugada-pattern ECG changes in the aforementioned ECG parameters at the baseline (Table 2). In the six male rabbits with LiCl (10 mmol/kg)-provoked Brugada-pattern ECG changes, one had bigeminal ventricular premature contractions (VPCs), one had complete atrioventricular block (AVB), and four (4/6, 83.3%) developed ventricular tachyarrhythmia after a long-short sequence (Figure 3). Among the 4 rabbits that developed ventricular arrhythmia, 3 rabbits had incessant ventricular fibrillation, while the other had sustained monomorphic ventricular tachycardia with an average ventricular rate of 8.3 Hz. In the male rabbit that developed VPCs, the frequency was 11 ± 1 VPCs/10 s. All the arrhythmic events in male rabbits were noted spontaneously without electrical stimulus between 5-30 min after LiCl (10 mmol/kg) administration.

### 3.2. Effects of LiCl on AP Morphology and Ionic Currents in RVOT Cardiomyocytes

There were no significant differences in the RMP, APA, APD_90_, APD_50,_ and APD_20_ between genders both at the baseline and after LiCl (100 μM) administration (Figure 4). Male RVOT cardiomyocytes had a larger I_Na_ than female RVOT cardiomyocytes. LiCl of 100 μM decreased the I_Na_ of RVOT cardiomyocytes in both male and female rabbits, and LiCl of 100 μM had greater I_Na_ inhibition in male RVOT cardiomyocytes than in female RVOT cardiomyocytes (Figure 5).

Moreover, male and female RVOT cardiomyocytes had a similar I_Na-late_. LiCl of 100 μM significantly increased the I_Na-late_ by 41.7% in male RVOT cardiomyocytes, but it did not change the I_Na-late_ in female RVOT cardiomyocytes (Figure 6). 

### 3.3. Effect of Ranolazine on the QRS Duration, QT Interval, and Electrical Activity in Male Rabbits

As shown in Figure 7, ranolazine 3.6 mg/kg did not change the QRS duration and QT interval but increased the QTc interval. In the presence of ranolazine, LiCl of 10 mmol/kg did not change the QRS duration, QT interval, or QTc interval. Compared to the previous inducibility of LiCl-induced Brugada-pattern ECG changes in male rabbits, LiCl of 10 mmol/kg did not provoke a Brugada-pattern ECG in the presence of ranolazine (6/9 versus 0/5, *p* = 0.031).

## 4. Discussion

In this study, for the first time, we successfully provoked Brugada-pattern ECG changes in male rabbits via the administration of high-dose LiCl. This finding is consistent with the known risk of lithium intoxication in humans. The simplicity of lithium administration to produce Brugada-pattern ECG changes and ventricular tachyarrhythmia in rabbits may provide a platform for mechanistic insights and pharmacological intervention in ventricular arrhythmogenesis during lithium intoxication. 

Syncope or cardiac arrest was mostly found in male patients with lithium-induced type 1 Brugada-pattern ECG changes [3,6,27]. In the current study, we also found lithium-induced Brugada-pattern ECG changes in only male rabbits. The more pronounced lithium (10 mmol/kg)-induced QRS prolongation in the male rabbits suggested that lithium had a greater impact on the male ventricular conduction property than on the female ventricular conduction property. The conduction delay, especially in the RVOT area, which can be reflected by surface ECG findings, may contribute to the Brugada syndrome phenotype [28,29]. The mutated cardiac Na^+^ channel causing discrete interstitial changes in RV with associated reentry and ventricular fibrillation originating from RVOT subendocardium has been documented [30]. Lithium’s interaction with Na^+^– Ca^2+^ exchangers and Na^+^/K^+^ pump could induce the changing physiology of the cellular membrane. Lithium can decrease the intracellular K^+^ concentration, replace intracellular Ca^2+^ and cause hypercalcemia, and induce various ECG changes [31]. Unmasking type 1 Brugada-pattern ECG changes by lithium may be caused by its blocking effects on the I_Na_ at subtherapeutic concentrations [1]. In the present study, we found that lithium differentially modulated the current densities of the I_Na_ and I_Na-late_ in male and female RVOT cardiomyocytes. These findings suggest that lithium exhibits sex differences in modulating the I_Na_ and I_Na-late_ of RVOT cardiomyocytes, which may explain the lithium-provoked Brugada-pattern ECG changes and ventricular tachyarrhythmias in the male rabbits in our study. The greater I_Na_ inhibition caused by 100 μM of lithium in male RVOT cardiomyocytes might have led to greater ECG changes and a greater incidence of Brugada-pattern ECG changes and tachyarrhythmia in male rabbits. The experiments in canine epicardial and endocardial layers show that male cardiomyocytes had a larger I_Na_ amplitude than female cardiomyocytes [32]. These findings suggest that sex differences in ventricular I_Na_ may potentially play a role in the higher risk of the Brugada syndrome in male individuals. Accordingly, we hypothesized that the male sex would increase the arrhythmogenic potential of Brugada syndrome via its greater impact on channelopathy in cardiomyocytes. The I_Na-late_, though a relatively small current relative to the peak I_Na_, was sufficiently large during the AP plateau to affect the AP duration. An enhanced I_Na-late_ in cardiomyocytes can lower the threshold of the AP, prolong the AP duration, and increase the intracellular Ca^2+^ concentration, which may initiate diastolic depolarization and increase excitability, leading to increased arrhythmogenesis [33]. However, the impact of gender on the relationship between lithium-induced Brugada-pattern ECG and the I_Na-late_ has not been specifically established [34]. We found that lithium increased the I_Na-late_ in only male RVOT cardiomyocytes. This finding indicates the impact of gender on the incidence of lithium-induced Brugada-pattern ECGs or ventricular tachyarrhythmia, suggesting that the I_Na-late_ may play a crucial role in the ventricular arrhythmogenesis of lithium intoxication. Gender-specific differences in I_Na-late_ have been reported in mice ventricular cells [19]. The augmentation of the I_Na-late_ by lithium is greater in males resulting in triggered activity caused by intracellular Ca^2+^ overload with the activation of the Na^+^/Ca^2+^ exchange [34]. Moreover, lithium provoked Ca^2+^-dependent receptor inactivation, which further accumulated intracellular Ca^2+^ by the inhibition of a Ca^2+^ extrusion via the Na^+^/Ca^2+^ exchange [35]. Although Ca^2+^ overload is usually related to a prolonged QTc, multiple case series have shown no significant changes in the QTc intervals of patients before and after initiating lithium therapy [36,37]. A previous study showed how ranolazine may reduce ventricular and atrial arrhythmias [38]. Within the therapeutic concentrations, ranolazine can inhibit the rapid component of a delayed rectifier potassium current (I_Kr_) and I_Na-late_. The inhibition of I_Kr_ by ranolazine prolongs AP duration, while its effect to inhibit I_Na-late_ abbreviates the AP duration. The clinical net effect of the inhibition of these ion channel currents sums up to a modest increase in the mean QTc interval over the therapeutic range of ranolazine [3]. In the current animal study, we found that ranolazine could abolish a lithium-induced longer QRS duration, Brugada-pattern ECG changes, and ventricular tachyarrhythmias, which suggests that I_Na-late_ inhibition may have a therapeutic implication for the management of lithium-induced ventricular arrhythmogenesis.

This study has some potential limitations. First, since the lithium dosage used in this study was relatively large, it is unclear whether our study may well correlate to lithium intoxication in clinical scenarios. Second, the anti-arrhythmogenic potential of ranolazine in lithium-induced Brugada-pattern ECG changes in rabbits might not be translated to the human Brugada syndrome, which is majorly caused by channelopathy. Moreover, our study just investigated the acute response of lithium on the Na^+^ current activity of RVOT cardiomyocytes. Theoretically, the histological presentation and the expression of proteins or mRNA levels would not be changed in a short period of time [39,40]. It is not clear whether the long-term treatment of lithium may have a different impact on RVOT channel activity. High-density mapping may help study the trigger of ventricular tachycardia in rabbits with lithium-induced Brugada-pattern ECG changes, although RVOT is widely believed to be the main arrhythmogenic area in clinical Brugada syndrome. Finally, the ECG lead placement on the anterior thorax of the rabbits has been adjusted to be different from that in humans during clinical practice, for the RV in rabbits is closer to the midline of the chest. 

## 5. Conclusions

Male sex is predisposed to lithium-induced Brugada-pattern ECG changes and ventricular conduction delay, with lithium having a greater impact on the I_Na_ and I_Na-late_ in male RVOT cardiomyocytes than in females (Figure 8). Targeting the I_Na-late_ may be a potential therapeutic strategy for Brugada syndrome-related ventricular tachyarrhythmia.

## Figures and Tables

**Figure 1 biomedicines-10-02727-f001:**
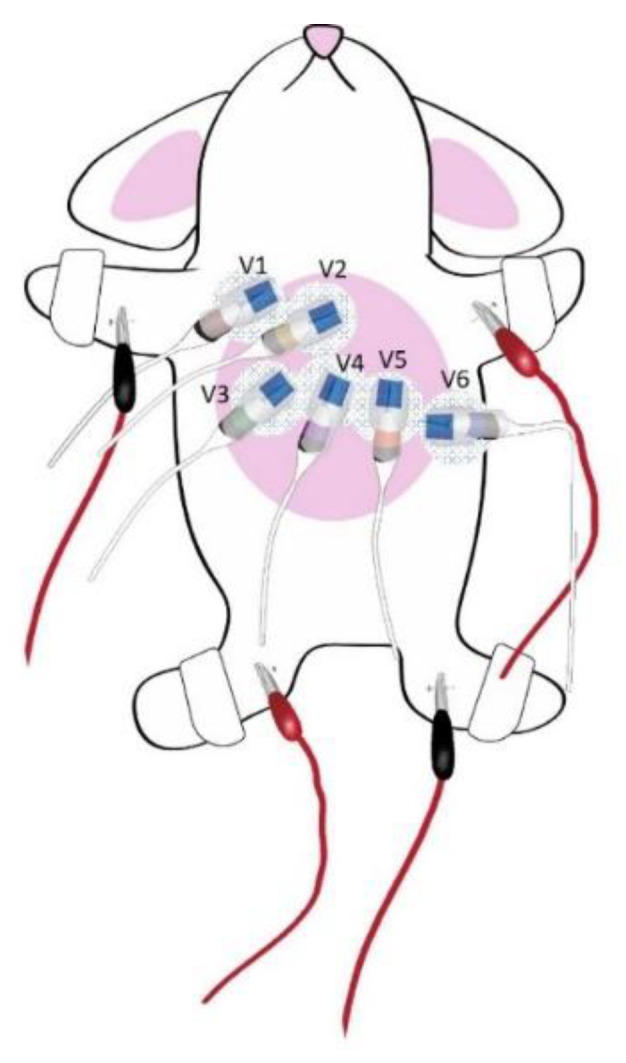
Schematic presentation of electrocardiographic recordings from the rabbits. The six precordial leads and limb leads were attached as illustrated to the anterior thorax of the rabbits. The locations of precordial leads were placed as follows-V2 and V4: in the 4th and 5th intercostal space on the sternum, respectively; V1 and V3: in the 4th and 5th intercostal space along the right margin of sternum, respectively; V6: in the 5th intercostal space along the mid-axillary line and V5 was mid-way between V4 and V6.

**Figure 2 biomedicines-10-02727-f002:**
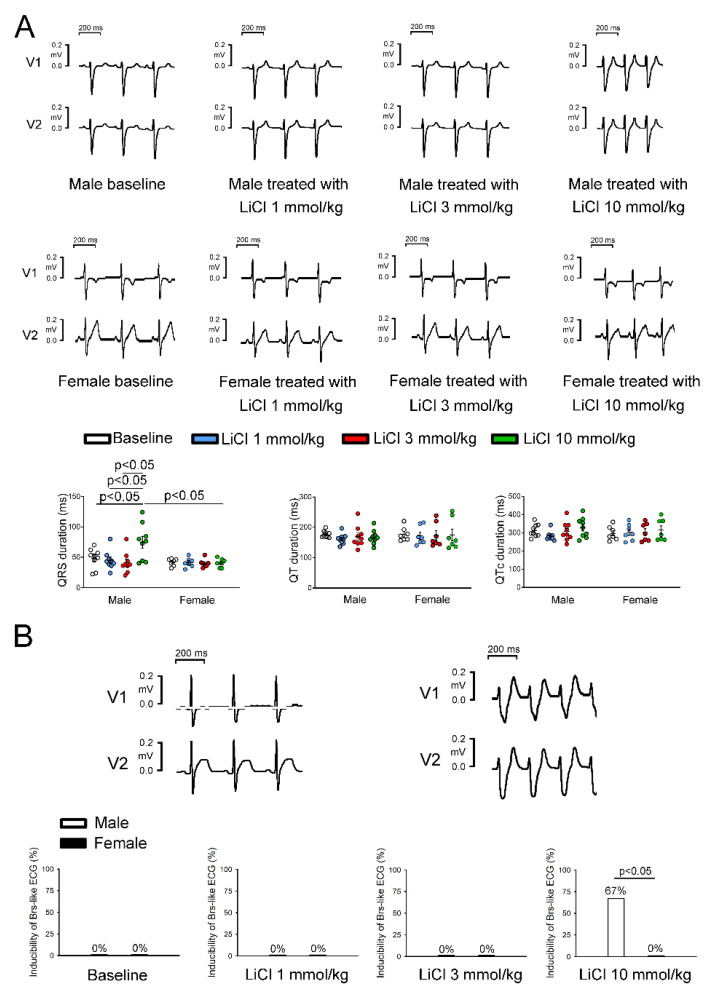
Electrocardiographic changes in lithium chloride (LiCl)-treated rabbits by sex. (**A**) Representative electrocardiographic tracings and average data of the QRS duration, QT interval, and corrected QT (QTc) interval from the male (*n* = 9) and female (*n* = 7) rabbits treated with 1, 3, and 10 mmol/kg of LiCl. (**B**) Representative electrocardiographic morphology during baseline and under 10 mmol/kg of LiCl. Listed on the bottom is the inducibility of Brugada-pattern electrocardiographic changes in male rabbits treated with 1, 3, and 10 mmol/kg of LiCl.

**Figure 3 biomedicines-10-02727-f003:**
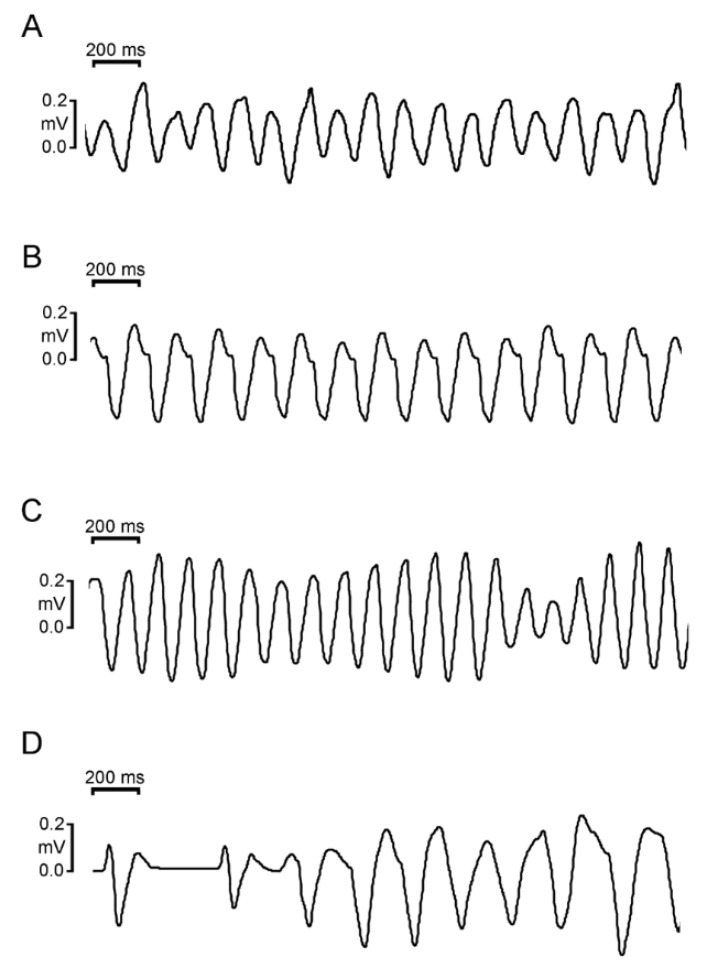
Development of various types of ventricular arrhythmias after the appearance of Brugada-pattern electrocardiographic (ECG) changes induced by lithium administration. The following representative ECG rhythm strips were recorded from lead II. (**A**) Ventricular fibrillation. (**B**) Ventricular tachycardia with right bundle branch block morphology. (**C**) Polymorphic ventricular tachycardia (**D**) Ventricular tachycardia occurred after a long-short sequence induced by the administration of 10.0 mmol/kg of lithium chloride in male rabbits.

**Figure 4 biomedicines-10-02727-f004:**
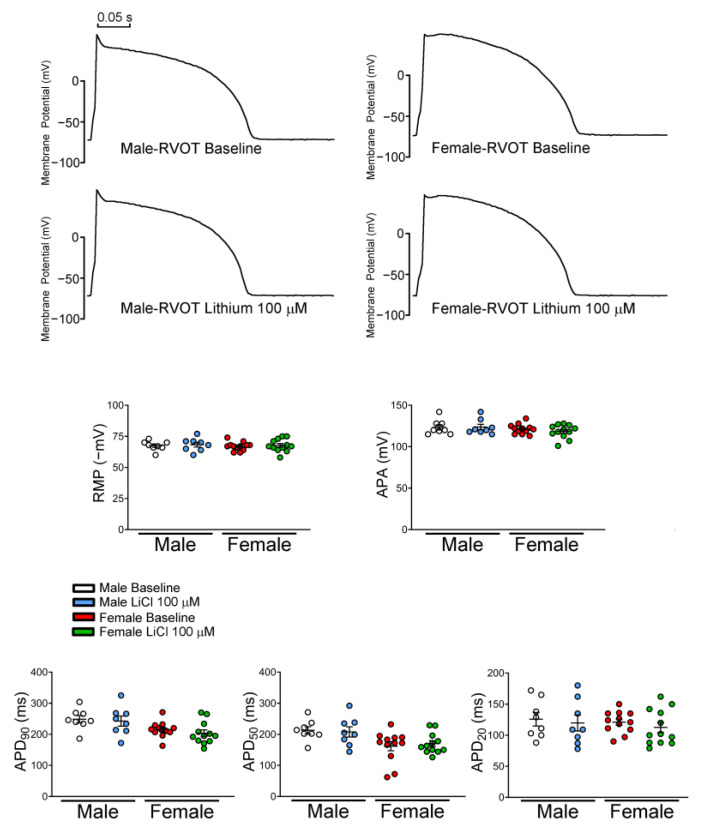
Effects of lithium on the electrophysiologic characteristics of rabbit right ventricular outflow tract (RVOT) cardiomyocytes. Representative tracings and average data of resting membrane potential (RMP), action potential amplitude (APA), and action potential duration at 90%, 50%, and 20% repolarization (APD_90,_ APD_50_, APD_20_) in male (*n* = 8) and female (*n* = 12) RVOT cardiomyocytes before and after lithium (100 μM) administration. LiCl = lithium chloride.

**Figure 5 biomedicines-10-02727-f005:**
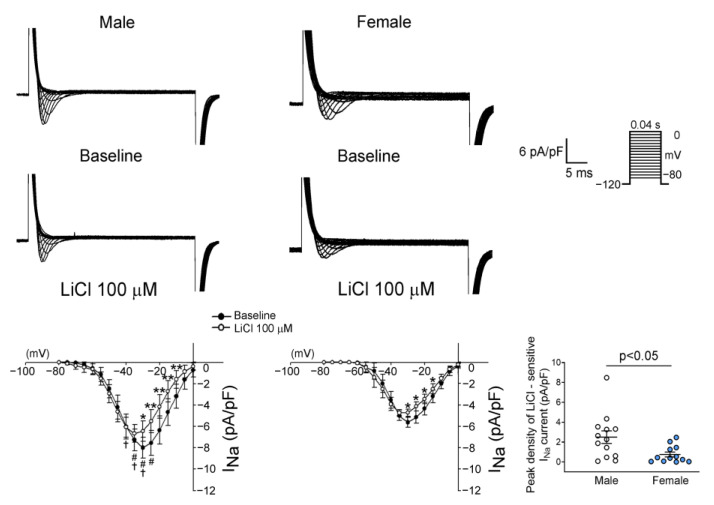
Effects of lithium on the sodium current (I_Na_) in right ventricular outflow tract (RVOT) cardiomyocytes. Representative tracings and I-V relationship of the sodium current (I_Na_) in male (*n* = 14) and female (*n* = 14) RVOT cardiomyocytes before and after lithium (100 μM) administration. There was greater lithium-sensitive I_Na_ in male RVOT cardiomyocytes than in female RVOT cardiomyocytes. * *p* < 0.05, ** *p* < 0.01, # *p* < 0.05 male versus female rabbits at baseline, ^†^
*p* < 0.05 male versus female rabbits after lithium (100 μM) administration.

**Figure 6 biomedicines-10-02727-f006:**
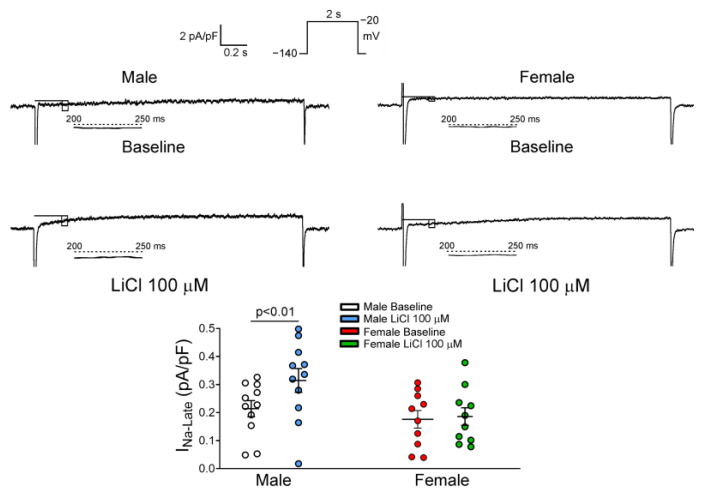
Effects of lithium on the late sodium current (I_Na-late_) in right ventricular outflow tract (RVOT) cardiomyocytes. Representative tracings and the difference of peak current density of the I_Na-late_ before and after lithium (100 μM) administration in male (*n* = 11) and female (*n* = 10) RVOT cardiomyocytes. Lithium significantly increased I_Na-late_ in male RVOT cardiomyocytes compared to female RVOT cardiomyocytes, LiCl = lithium chloride.

**Figure 7 biomedicines-10-02727-f007:**
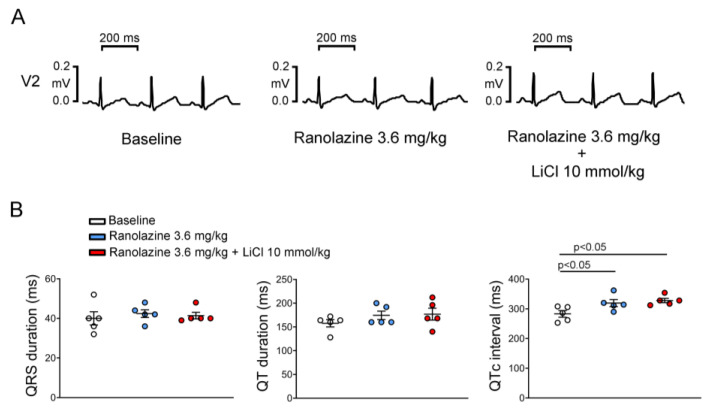
Effects of ranolazine on electrocardiography parameters in lithium chloride (LiCl)-treated male rabbits. Representative electrocardiography tracings (**A**) and average data (**B**) of the QRS duration, QT interval, and corrected QT (QTc) interval from male rabbits (*n* = 5) treated with LiCl (10 mmol/kg) with ranolazine pretreatment.

**Figure 8 biomedicines-10-02727-f008:**
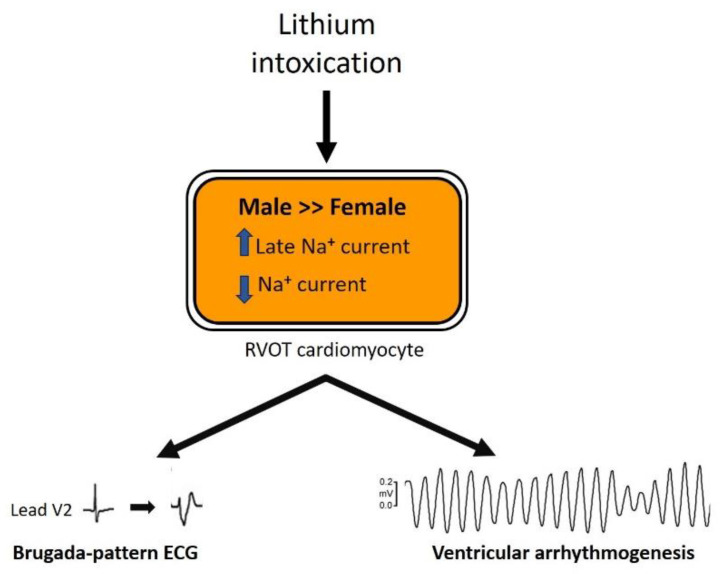
The underlying mechanisms driving the differences in lithium-induced electrical changes and arrhythmogenicity between genders. The up arrow represents an increase in current density; the down arrow represents a decrease in current density. ECG, electrocardiography; Na, sodium; RVOT, right ventricular outflow tract.

**Table 1 biomedicines-10-02727-t001:** ECG parameters before and after intravenous lithium administration at different concentrations.

ECG Parameters	Baseline			1.0 mmol/kg			3.0 mmol/kg			10.0 mmol/kg		
	Male	Female	*p* Value	Male	Female	*p* Value	Male	Female	*p* Value	Male	Female	*p* Value
	*n* = 9	*n* = 7		*n* = 9	*n* = 7		*n* = 9	*n* = 7		*n* = 9	*n* = 7	
**HR (Hz)**	3.1 ± 0.4	2.8 ± 0.4	0.247	3.0 ± 0.2	2.6 ± 0.3	0.272	3.1 ± 0.3	2.9 ± 0.7	0.392	3.6 ± 0.6	3.5 ± 0.9	0.810
**P-R interval (ms)**	69.4 ± 6.5	71.1 ± 9.4	0.691	60.0 ± 0.1	78.2 ± 15.6	0.181	68.6 ± 10.3	67.9 ± 8.6	0.909	59.8 ± 17.0	65.8 ± 24.3	0.614
**QRS duration (ms)**	48.3 ± 16.1	41.7 ± 6.1	0.284	30.0 ± 14.1	39.6 ± 11.3	0.513	40.9 ± 10.3	36.5 ± 13.4	0.604	74.7 ± 33.9	40.7 ± 7.0 *	0.023
**R wave amplitude (mV)**	0.22 ± 0.14	0.23 ± 0.16	0.896	0.33 ± 0.25	0.35 ± 0.12	0.900	0.21 ± 0.16	0.26 ± 0.12	0.616	0.11 ± 0.12	0.16 ± 0.08	0.368
**QT interval (ms)**	176.7 ± 11.2	176.8 ± 22.9	0.993	184.0 ± 17.0	180.9 ± 30.6	0.893	174.6 ± 43.1	185.2 ± 45.4	0.733	167.4 ± 26.8	175.2 ± 50.4	0.728
**QTc (ms)**	311.7 ± 34.5	296.6 ± 38.3	0.430	318.0 ± 42.4	290.7 ± 28.9	0.525	306.4 ± 61.8	307.3 ± 53.1	0.983	314.1 ± 48.4	315.0 ± 59.1	0.977
**T wave amplitude (mV)**	0.08 ± 0.05	0.11 ± 0.05	0.186	0.10 ± 0.01	0.12 ± 0.07	0.617	0.08 ± 0.03	0.11 ± 0.06	0.359	0.08 ± 0.06	0.10 ± 0.09	0.579

The QRS duration at 10 mmol/kg LiCl was significantly longer in male rabbits than in female rabbits. * *p* < 0.05 vs. males at 10 mmol/kg LiCl. Values are expressed as mean ± standard error of the mean. ECG = electrocardiography; HR = heart rate; LiCl = lithium chloride; QTc = corrected QT interval.

**Table 2 biomedicines-10-02727-t002:** Baseline ECG parameters in male rabbits with and without provokable Brugada-pattern ECG findings.

Baseline Parameters	Provokable	Non-Provokable	*p*-Value
	(*n* = 6)	(*n* = 3)	
**HR (Hz)**	3.3 ± 0.2	2.8 ± 0.2	0.151
**P-R interval (ms)**	67.0 ± 2.5	74.1 ± 3.0	0.130
**QRS duration (ms)**	51.5 ± 16.8	41.8 ± 9.0	0.429
**R wave amplitude (mV)**	0.19 ± 0.03	0.27 ± 0.15	0.662
**R+S amplitude (mV)**	0.26 ± 0.04	0.33 ± 0.14	0.541
**QT interval (ms)**	178.4 ± 5.4	173.3 ± 3.4	0.561
**QTc (ms)**	322.2 ± 13.9	290.7 ± 16.9	0.217
**T wave amplitude (mV)**	0.07 ± 0.01	0.11 ± 0.04	0.231

Values are expressed as mean ± standard error of the mean. ECG = electrocardiography; HR = heart rate; QTc = corrected QT interval.

## Data Availability

All data generated or analyzed during the current study are included in this published article and are available from the corresponding author upon reasonable request.

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
