# Peer review of "Gender Difference in Lithium-Induced Sodium Current Dysregulation and Ventricular Arrhythmogenesis in Right Ventricular Outflow Tract Cardiomyocytes"

_biomedicines, 2022, doi:10.3390/biomedicines10112727_

Round 1
Reviewer 1 Report
I congratulate with the Authors for this interesting and original experimental article, with potential clinical relevance in the understanding of the pathogenetic mechanism of Brugada Syndrome.
My suggestions for this manuscript are the following:
1) In the Introduction (line 71), you could quote a recent study, showing that new ECG analysis enabled the localization of the electrical substrate of the Brugada pattern to the right ventricle outflow tract (doi: 10.1161/CIRCEP.120.008524). Also, (line 78), you should add the important concept that Brugada Syndrome phenotype can be eliminated by epicardial substrate ablation in Brugada patients (doi: 10.1161/CIRCEP.115.003220).
2) In the Results, you should provide a new Table 1, including Baseline ECG Parameters in Males vs. Females rabbits at different Lithium concentrations (in an extended format similar to current Table 1). Viceversa current Table 1 should become Table 2 (note that should you have some space limitations, this Table could be eliminated, since it contains limited information, without showing significant differences).
Author Response
Responses to Reviewer #1
Thank you for your detailed comments, which are very helpful to improve the manuscript. Our responses to these comments are below and the relevant passages have been incorporated into the revised manuscript and marked in red font.
- Regarding the comment “In the Introduction (line 71), you could quote a recent study, showing that new ECG analysis enabled the localization of the electrical substrate of the Brugada pattern to the right ventricle outflow tract (doi: 10.1161/CIRCEP.120.008524). Also, (line 78), you should add the important concept that Brugada Syndrome phenotype can be eliminated by epicardial substrate ablation in Brugada patients (doi: 10.1161/CIRCEP.115.003220)”
Response 1: Thank you for this very insightful comment. According to your suggestion, we have quoted the study (reference 12) to highlight RVOT as the arrhythmogenic substrate of Brugada syndrome proven by a new method of ECG analysis, and also commented the important concept that Brugada syndrome phenotype can be eliminated by epicardial substrate ablation in Brugada syndrome patients with a new reference (reference 16) (page 2, line 73-75; page 2, line 81-85) as follows “The abnormal ECG activation in Brugada syndrome can be temporo-spatially correlated to RVOT by a new CineECG method.”, and “In patients with Brugada syndrome, low-voltage areas or area with abnormal electrogram were commonly identified on the RVOT. Epicardial ablation of provoked substrate in the RVOT in Brugada patients can eliminate Brugada syndrome phenotype, which is associated with freedom from ventricular arrhythmia inducibility. - Regarding the comment “In the Results, you should provide a new Table 1, including Baseline ECG Parameters in Males vs. Females rabbits at different Lithium concentrations (in an extended format similar to current Table 1). Vice versa current Table 1 should become Table 2 (note that should you have some space limitations, this Table could be eliminated, since it contains limited information, without showing significant differences)”
Response 2: Thank you very much for your comment. According to your suggestion, we have added a new table 1 in the revised results, which provides ECG parameters during baseline as well as at different lithium concentrations in male vs. female rabbits. As disclosed in table 1, there were no significant differences in the heart rate, QT interval, corrected QT (QTc) interval, P wave amplitude, or T wave amplitude between genders during baseline, as well as after intravenous administration lithium chloride (1 and 3 mmol/kg). After administration of lithium chloride (10 mmol/kg), there was significantly longer QRS duration in male as compared to female rabbits. The original table 1 was changed to table 2.
The above descriptions are the responses to your comments and suggestions.
Sincerely yours,
Yi-Jen Chen, MD, PhD

Reviewer 2 Report
In this manuscript, Liu et al. correlates the lithium intoxication and its ECG induced changes to changes in INa and INa,Late sodium current densities. The authors propose targeting INa,Late as a therapeutic strategy in Brugada induced ventricular arrhythmias. Some concerns are noted and need to be addressed to improve the overall impact of the study.
1- First line in the abstract, the mechanism underlying the predisposition of male gender in symptomatic Brugada syndrome is not clear. I don't think the presented data address this question. There are no mechanistic studies to explain the electrical findings.
2- Did the authors look at the Nav1.5 protein expression in male myocytes/hearts post Lithium treatment? this data can provide an initial mechanistic explanation of the changes noted in INa and INa,Late.
3- The authors reported changes in INa,Late and referred it to testosterone levels. This information contradicts Reference 16 cited by the authors (Lowe et al. 2012) as the changes in INa,Late were previously linked to female mice and female sex hormones.
4- Why did the authors call the ECG changes noted after Lithium, Brugada like syndrome, if no changes in QTc and T-wave dimensions were reported post Lithium in male rabbits?
5- The authors displayed the arrhythmic events as number of rabbits who displayed arrhythmias, have they counted the number of individual events post lithium in each rabbit?
6- Any data on APD changes in the rabbit's myocytes post Lithium to show if the APD90 is prolonged post Lithium?
Author Response
Thank you for your detailed comments, which are very helpful to improve the manuscript. Our responses to these comments are below and the relevant passages have been incorporated into the revised manuscript and marked in red font.
1. Regarding the comment “First line in the abstract, the mechanism underlying the predisposition of male gender in symptomatic Brugada syndrome is not clear. I don't think the presented data address this question. There are no mechanistic studies to explain the electrical findings.”
Response 1: We appreciate your comment very much. In the present study, we did not provide mechanistic studies to explain lithium-induced electrical findings in male rabbit, and we deleted this sentence from the abstract according to your suggestion.
2. Regarding the specific comment “Did the authors look at the Nav1.5 protein expression in male myocytes/hearts post Lithium treatment? this data can provide an initial mechanistic explanation of the changes noted in INa and INa, Late.”
Response 2: Thank you for this insightful comment. According to the previous study (Yanagita et al., Neuropharmacology 2007), acute treatment with lithium chloride did not change the expressions of ionic channel protein or mRNA, while only chronic lithium treatment could up-regulate cell surface Na+ channel. Theoretically, the quantitative expression analysis is not changed due to the short (within minutes) administration of lithium chloride, and thus we did not perform western blot in the current experimental setting. We have added this limitation in the revised discussion section as follows (page 11, line 338; page 12, line 339-342)” Moreover, our study just investigated acute response of lithium on Na+ current activity of RVOT cardiomyocytes. Theoretically, the histological presentation, and the expression of protein or mRNA level would not be changed in a short period of time (Yanagita et al., Neuropharmacology 2007; Gong R et al., Am J Physiol Renal Physiol. 2016). It is not clear whether long-term treatment of lithium may have different impact on RVOT channel activity.”
3. Regarding the comment “The authors reported changes in INa,Late and referred it to testosterone levels. This information contradicts Reference 16 cited by the authors (Lowe et al. 2012) as the changes in INa,Late were previously linked to female mice and female sex hormones.”
Response 3: Thank you very much for this helpful comment. We understand your concern that we had mentioned about male predominance in lithium-induced Brugada-pattern ECG and INa-late changes may be owing to a higher testosterone level in the Brugada syndrome phenotype, which is not compatible with the results of Lowe’s work which showed a larger INa-late in female ventricular myocytes. Because we did not further investigate the effect of testosterone on lithium-induced Brugada-pattern ECG and INa-late changes, we cannot confirm that male predominance of lithium-induced changes is related with testosterone. Accordingly, we deleted the description “A higher testosterone level may have a significant role in the Brugada syndrome phenotype.” from the discussion section of revised manuscript.
4. Regarding the comment “Why did the authors call the ECG changes noted after Lithium, Brugada like syndrome, if no changes in QTc and T-wave dimensions were reported post Lithium in male rabbits?”
Response 4: We appreciate this comment very much. In our study, we defined Brugada-pattern ECG change as cove-shaped or saddle-back ST segment elevation >2 mm in the right precordial leads and ≧0.5 mm elevation of the terminal ST segment with a positive or biphasic T wave as mentioned in the methods. The change of QTc or T wave dimension are not involved in the typical Brugada ECG pattern, and there is no significant change in the QTc intervals of rabbits before and after initiating lithium therapy described in the discussion (Sibarov DA. J Pharmacol Exp Ther. 2015 and Bucht G. Acta Med Scand. 1984)
5. Regarding the comment “The authors displayed the arrhythmic events as number of rabbits who displayed arrhythmias, have they counted the number of individual events post lithium in each rabbit?”
Response 5: Thank you very much for this insightful comment. According to your suggestion, we have described the number of arrhythmic events in the revised results as follows (page 5, line 196-200)” Among the 4 rabbits that developed ventricular arrhythmia, 3 rabbits had incessant ventricular fibrillation, while the other had sustained monomorphic ventricular tachycardia with average ventricular rate 8.3 Hz. In the male rabbit that developed VPCs, the frequency was 11 ± 1 VPCs / 10 seconds.”
6. Regarding the comment “Any data on APD changes in the rabbit's myocytes post Lithium to show if the APD90 is prolonged post Lithium?”
Response 6: Thank you very much for this helpful comment. According to your suggestion, we have analyzed the effects of LiCl on APD in RVOT cardiomyocytes, as illustrated in the new figure 4 (the original figure 4 was changed to figure 5). We found that LiCl (100 μM) did not change the APD in male or female RVOT cardiomyocytes and there were no significant differences in RMP, APA, APD90, APD50 and APD20 between genders both at baseline and after LiCl (100 μM) administration. We have provided these results in the revised Figure 4 and results section (page 7, line 228-229): “There were no significant differences in RMP, APA, APD90, APD50 and APD20 between genders both at baseline and after LiCl (100μM) administration.”, and added the methods for measuring transmembrane potentials in the revised methods section (page 4, line 152-161).
The above descriptions are the responses to your comments and suggestions.
Sincerely yours,
Yi-Jen Chen, MD, PhD

Reviewer 3 Report
The paper titled: <> is authored by Ching-Han Liu et al
Although an interesting approach, some concerns could be addressed to improve the demonstration.
1- Transform all bar graphs into scatter/dot-plot graphs showing means and error bars.
2- Provide a Highlight section with up to 5-bullet points to describe the main discoveries.
3- A major weakness in this paper is that Ventricular arrhythmias are described in Figure 3, but no details are provided about how they were characterized.
Were they spontanous arrhythmias generated after the administration of lithium?
4-How long after the administration of lithium the Ventricular arrhythmias occured?
Were they induced by electrical stimulation of the heart ? If yes, which catheter, which voltage etc
5- Provide a Schematic to summarize your discoveries
6- Why no biomolecular analyses were performed, including qPCR or western blot ?
7-Also, I believe that some histological analyses could improve your demonstration
Author Response
Thank you for your detailed comments, which are very helpful to improve the manuscript. Our responses to these comments are below and the relevant passages have been incorporated into the revised manuscript and marked in red font.
1. Regarding the comment “Transform all bar graphs into scatter/dot-plot graphs showing means and error bars.”
Response 1: We are thankful to your precious comment. As you suggested, we have transformed all the bar graphs in the figures into dot-plot graphs with means and error bars, if applicable.
2. Regarding the specific comment “Provide a Highlight section with up to 5-bullet points to describe the main discoveries.”
Response 2: Thank you very much for this helpful comment. As you suggested, we have listed 4 bullet points following the conclusions as follows (page 12, line 358-368):
- High-dose LiCl provokes Brugada-pattern ECG changes in male rabbits, and those were then turned highly susceptible to ventricular tachyarrhythmias.
- Lithium inhibits the INa and increases the INa-late only in male RVOT cardiomyocytes, which causes greater ventricular conduction delay and RVOT arrhythmogenesis.
- Male predominance in lithium-induced Brugada-pattern ECG and INa-late changes suggests that male sex would increase the arrhythmogenic potential of lithium intoxication via its greater impact on Na+ channelopathy in male cardiomyocytes.
- Ranolazine, an inhibitor of INa-late inhibitor, abolished the conduction delay and fatal arrhythmia imposed by lithium, thus offering a potential therapeutic target for Brugada syndrome.
3. Regarding the comment “A major weakness in this paper is that Ventricular arrhythmias are described in Figure 3, but no details are provided about how they were characterized.”
Response 3: Thank you very much for your insightful suggestions. We are sorry for our vague description. According to your suggestion, we have added more detailed descriptions in the revised results section as follows (page 5, line 196-201)” Among the 4 rabbits that developed ventricular arrhythmia, 3 rabbits had incessant ventricular fibrillation, while the other had sustained monomorphic ventricular tachycardia with average ventricular rate 8.3 Hz. In the male rabbit that developed VPCs, the frequency was 11 ± 1 VPCs / 10 seconds. All the arrhythmic events in male rabbits were noted spontaneously without electrical stimulus between 5-30 minutes after LiCl (10 mmol/kg) administration.”
4. Regarding the comment “How long after the administration of lithium the Ventricular arrhythmias occurred? Were they induced by electrical stimulation of the heart? If yes, which catheter, which voltage etc.”
Response 4: Thank you very much for your comment. We are sorry for our unclear presentation in this manuscript. We have added a more detailed description in the revised results section as follows (page 5, line 200-201)” All the arrhythmic events in male rabbits were noted spontaneously without electrical stimulus between 5-30 minutes after LiCl (10 mmol/kg) administration.”
5. Regarding the comment “Provide a Schematic to summarize your discoveries”
Response 5: Thank you very much for this comment. According to your suggestion, we have added a schematic figure (revised Figure 8) following conclusions for graphical highlights of the main findings.
6. Regarding the comment “Why no biomolecular analyses were performed, including qPCR or western blot?”
Response 6: Thank you for this insightful comment. According to the previous study (Yanagita et al., Neuropharmacology 2007), acute treatment with lithium chloride did not change the expressions ionic channel protein, while only chronic lithium treatment could up-regulate cell surface Na+ channel. Theoretically, the quantitative expression analysis is not changed due to the short (within minutes) administration of lithium chloride, and thus we did not perform qPCR or western blot in the current experimental setting. We have added this limitation in the revised discussion section as follows (page 11, line 338; page 12, line 339-342)” Moreover, our study just investigated acute response of lithium on Na+ current activity of RVOT cardiomyocytes. Theoretically, the histological presentation, and the expression of protein or mRNA level would not be changed in a short period of time (Yanagita et al., Neuropharmacology 2007; Gong R et al., Am J Physiol Renal Physiol. 2016). It is not clear whether long-term treatment of lithium may have different impact on RVOT channel activity.”
7. Regarding the comment “Also, I believe that some histological analyses could improve your demonstration.”
Response 7: Thank you very much for this comment. Lithium was clinically reported to cause organ damage mostly after days to weeks of administration (Gong R et al., Am J Physiol Renal Physiol. 2016). In our study, the duration of lithium exposure lasted only up to 2 hours in each rabbit, during which the effect of lithium was probably limited to the molecular and cellular levels. We explain the reasons and describe them in the limitation (page 11, line 338; page 12, line 339-342).
The above descriptions are the responses to your comments and suggestions.
Sincerely yours,
Yi-Jen Chen, MD, PhD

Round 2
Reviewer 2 Report
The authors were very responsive to my previous concerns by either adding more data and/or editing the text.
Author Response
Thank you for your detailed comment, which is very helpful to improve the manuscript. Our response to this comment is below.
- Regarding the general comment “English language and style are fine/minor spell check required”
Response: We are very much thankful to you for your deep and thorough review and are grateful for your positive and encouraging comments. According to your suggestion, we have checked our revised manuscript's spelling and style in detail.
The above descriptions are the responses to your comments and suggestions.
Sincerely yours,
Yi-Jen Chen, MD, PhD
